# Schistosome eggs stimulate reactive oxygen species production to enhance M2 macrophage differentiation and promote hepatic pathology in schistosomiasis

**Yanxiong Yu**[1]ʘ, **Junling Wang**[1]ʘ, **Xiaohong Wang**[1], **Pan Gu**[1], **Zhigang Lei**[1], **Rui Tang**[1], **Chuan Wei**[1], **Lei Xu**[1], **Chun Wang**[1], **Ying Chen**[1], **Yanan Pu**[1], **Xin Qi**[1], **Beibei Yu**[1], **Xiaojun Chen**[1], **Jifeng Zhu**[1], **Yalin Li**[1], **Zhijie Zhang**[2], **Sha Zhou**[1]*, **Chuan Su**[1]*

**1** Department of Pathogen Biology and Immunology, Jiangsu Key Laboratory of Pathogen Biology, Center for Global Health, Nanjing Medical University, Nanjing, Jiangsu, China, **2** Department of Epidemiology and Biostatistics, School of Public Health, Fudan University, Shanghai, China

ʘ These authors contributed equally to this work.
* shazhou@njmu.edu.cn (SZ); chuansu@njmu.edu.cn (CS)

**Data Availability Statement:** All relevant data are within the manuscript.

## Abstract

Schistosomiasis is a neglected tropical disease of public health concern. The most devastating pathology in schistosomiasis japonica and mansoni is mainly attributed to the egg-induced granulomatous response and secondary fibrosis in host liver, which may lead to portal hypertension or even death of the host. Schistosome eggs induce M2 macrophages-rich granulomas and these M2 macrophages play critical roles in the maintenance of granuloma and subsequent fibrosis. Reactive oxygen species (ROS), which are highly produced by stimulated macrophages during infection and necessary for the differentiation of M2 macrophages, are massively distributed around deposited eggs in the liver. However, whether ROS are induced by schistosome eggs to subsequently promote M2 macrophage differentiation, and the possible underlying mechanisms as well, remain to be clarified during *S. japonicum* infection. Herein, we observed that extensive expression of ROS in the liver of *S. japonicum*-infected mice. Injection of ROS inhibitor in infected mice resulted in reduced hepatic granulomatous responses and fibrosis. Further investigations revealed that inhibition of ROS production in *S. japonicum*-infected mice reduces the differentiation of M2, accompanied by increased M1 macrophage differentiation. Finally, we proved that *S. japonicum* egg antigens (SEA) induce a high level of ROS production via both nicotinamide adenine dinucleotide phosphate (NADPH) oxidase 2 (NOX2) and mitochondria in macrophages. Our study may help to better understand the mechanism of schistosomiasis japonica-induced hepatic pathology and contribute to the development of potential therapeutic strategies by interfering with ROS production.

**Funding:** This work was supported by grants from the National Key Research and Development Program of China (MOST No: 2018YFA0507300) and the National Natural Science Foundation of China (NSFC No: 82030061 and No: 81871675) to C.S, the Natural Science Foundation of Jiangsu Province (No. BK20170105) and the National Natural Science Foundation of China (NSFC No. 81971963) to S.Z. The funders had no role in study design, data collection and analysis, decision to publish, or preparation of the manuscript.

**Competing interests:** The authors have declared that no competing interests exist.

## Author summary

Schistosomiasis is a neglected parasitic disease of poverty that affects ~200 million people mainly in (sub)tropical regions, resulting in a massive health burden and serious morbidity. During *Schistosoma japonicum* (*S. japonicum*) or *S. mansoni* infection, parasite eggs are trapped in host liver and induce hepatic granulomas and fibrosis, which leads to severe liver damage, and even death of the host. In hepatic schistosomiasis, considerable amounts of ROS accumulate in granulomas surrounding liver-trapped eggs. However, whether schistosome eggs trigger the production of ROS, and if so, whether and how ROS promote hepatic pathology in host remain unknown. In this study, the authors illustrated that *S. japonicum* eggs evoke high production of ROS in macrophages, which is necessary for egg-mediated M2 macrophage differentiation and promotes hepatic granulomas and fibrosis in *S. japonicum*-infected mice. These discoveries provide a potential target regarding schistosome eggs-induced ROS production, which can be manipulated to regulate immunopathology in hepatic schistosomiasis.

## Introduction

Schistosomiasis is a neglected tropical disease of poverty that impacts ~200 million people worldwide, resulting in a massive health burden and substantial morbidity. *Schistosoma mansoni* (*S. mansoni*), *S. japonicum*, and *S. haematobium* are schistosome species of major medical importance [1,2]. In *S. mansoni* and *S. japonicum* infections, the eggs produced by adult worms are the leading cause of liver pathology. The granulomas form around the trapped eggs throughout the host liver. The long-lasting egg-induced granulomatous inflammation may eventually cause cirrhosis, portal hypertension, and even liver failure and death of the host [3].

During schistosome infection, macrophages, one of the major cell components of hepatic granuloma and play important roles in both innate and adaptive immunity, are known to regulate the initiation, maintenance, and resolution of chronic granulomatous inflammation depending on their effector phenotypes [4]. Macrophages are functionally activated and polarized into either M1 or M2 phenotypes (called M1 or M2 macrophages, respectively) when responding to diverse stimuli. M1 macrophages are shown to be cytotoxic to schistosomula and also play a role in preventing hepatic fibrosis [4]. However, schistosome eggs induce M2 macrophage-rich granulomas, in which M2 macrophages enhance Th2-biased responses and prevent acute mortality but promote the granulomatous and fibrotic development in the liver during the chronic stage [4–6]. Our previous study has shown that *S. japonicum* egg antigens (SEA) preferentially promote M2 macrophage differentiation [7]. However, we still know very little about the exact mechanisms by which SEA preferentially promote M2 macrophage differentiation.

Reactive oxygen species (ROS) are a group of highly reactive chemicals containing oxygen and involved in various disorders and biological functions. Besides, ROS are also critical elements for innate and adaptive immune response [8–10]. As professional phagocytes, macrophages produce large amounts of ROS as their main weapon against invading pathogens and are one of the main cell sources of ROS production during hepatic fibrosis [10–12]. Critically, large amounts of reactive oxygen intermediates are produced in M2 macrophage-rich granulomas and this phenomenon reached its maximum intensity close to *S. mansoni* eggs [13]. Recently, researchers have revealed that ROS are necessary for M2 but not M1 macrophage differentiation [14,15]. However, whether schistosome eggs stimulate macrophages to produce more ROS in hepatic granulomas remains undefined, and if so, whether it subsequently

contributes to the increased M2 macrophage differentiation and thereby promotes granulomatous and fibrotic development in the liver also needs to be resolved.

Here, we aimed to explore the mechanisms of induction of ROS and its role in hepatic macrophage differentiation and development of liver immunopathology during *S. japonicum* infection. We found that schistosome egg antigens stimulate a high level of ROS production in macrophages, and ROS enhance M2 macrophage differentiation and promote hepatic pathology in mice with schistosomiasis japonica. Thus, the regulation of ROS production is suggested as a promising strategy for control of liver immunopathology of hepatic schistosomiasis.

## Materials and methods

### Ethics statement

All the animal experiments were performed in strict accordance with the Regulations for the Administration of Affairs Concerning Experimental Animals. All animals were handled with the approval by the Institutional Animal Care and Use Committee (IACUC) of Nanjing Medical University (Approval Number: IACUC-1905013).

### Mice and treatment

Specific pathogen-free (SPF) 8-week-old male BALB/c mice were purchased from the Animal Center of Nanjing Medical University (Nanjing, China). All mice were bred in an SPF breeding facility with standard environment conditions (room temperature, 22˚C; humidity, 40%; 12-h light-dark cycle). All efforts were made to minimize suffering.

BALB/c mice were infected percutaneously by exposure of the abdominal skin with 12 cercariae of the Chinese mainland strain of *S. japonicum* from infected snails (*Oncomelania hupensis*), which were acquired from the Jiangsu Institute of Parasitic Diseases (Wuxi, China). Infected mice were randomly chosen for treating with an NADPH oxidase 2 (NOX2) inhibitor, apocynin (Apo, 2 mg/mouse per day; Sigma-Aldrich, St. Louis, MO), or saline orally every day for 4 weeks, starting 4 weeks post-infection. At 8 weeks post-infection, all the mice were sacrificed for further study.

### Liver pathology

The right lower lobe of the liver was harvested from infected mice and fixed in 4% buffered formalin, embedded in paraffin, and then sectioned at 5 to 7 μm. The liver sections were stained with hematoxylin and eosin (H&E) to reveal granulomas. All the histopathological images were captured at 100× magnification using Axiovert 200M microscope coupled with a digital camera (Zeiss, Oberkochen, Germany).

The liver sections were stained using the Masson's Trichrome staining kit (Sigma-Aldrich) for semi-quantitative analysis of hepatic fibrosis. Six to eight fields from each slide were randomly obtained at 100× magnification using Axiovert 200M microscope coupled with a digital camera (Zeiss). The blue-stained area per total area and the fibrosis intensity were determined with Image-Pro Plus software (Media Cybernetics, Rockville, MD). The total fibrosis density score was determined by dividing the image intensity by the image area. Intensity exclusion parameters were identical for all the captured images.

### ROS production measurement in mouse liver

Frozen liver tissues were sectioned into 10-μm slices and washed with ice-cold PBS. Then the slices were soaked in 10 μM DCFH-DA (2′,7′-dichlorofluorescein-diacetate; Invitrogen,

Carlsbad, CA) for 30 min in the dark, followed by 3× wash in PBS. The slices were further counterstained with DAPI for better observation. Fluorescence intensity (488 nm excitation/ 525 nm emission) was determined with the fluorescence microscope at 200× magnification (Zeiss).

## Preparation of soluble egg antigens (SEA)

SEA were prepared as previously described [16,17]. Antigens were extracted from isolated eggs using sonication. The supernatant was recovered by centrifugation and filter-sterilized with a 0.22 μm filter and the endotoxin was removed with Polymyxin B-Agarose (Sigma-Aldrich). The remaining endotoxin activity (<0.01 EU/μg) was determined using the LAL assay kit (ThermoFisher Scientific, MA, USA). Protein concentration was determined using the DC Protein Assay Kit (Bio-Rad, Hercules, CA).

## Cell line and cell culture

The mouse macrophage cell line J774A.1 (American Type Culture Collection, Manassas, VA) were cultured in Dulbecco's Modified Eagle's medium (DMEM, Gibco, Grand Island, NY) with 10% (v/v) heat-inactivated fetal bovine serum (FBS, Gibco) and 1% (v/v) antibiotics (penicillin and streptomycin, Gibco).

J774A.1 cells were pretreated for an hour with or without Apo (100 μM) and then cultured for 24 hours in the absence or presence of SEA (40 μg/ml). Thereafter, cells were collected for further analysis.

## Serum ALT/AST analysis

The levels of serum alanine transaminase (ALT) and aspartate transaminase (AST) were tested by Olympus AU2700 Chemical Analyzer (Olympus, Tokyo, Japan) according to the manufacturer's guide.

## Flow cytometry (FCM) analysis

Mouse hepatic mononuclear cells (MNCs) were isolated using Percoll density-gradient centrifugation as described previously [18]. To analyze CD16/32 and CD206 expression on macrophages, MNCs were surface stained with anti-F4/80-PerCP-Cy5.5 (eBioscience, San Diego, CA), anti-16/32-PE (eBioscience), and anti-CD206-APC (R&D Systems, Minneapolis, MN).

To determine Th1, Th2, and Th17 cells, splenocytes were stimulated with ionomycin (1 μg/ ml, Sigma-Aldrich) and phorbol myristate acetate (PMA, 25 ng/ml, Sigma-Aldrich) in the presence of 1 μl/ml of Golgistop (BD Bioscience, San Diego, CA). After 6 hours, the cells were collected and surface stained with anti-CD3e-PerCP-Cy5.5 (eBioscience) and anti-CD4-FITC (eBioscience), and washed, fixed, and permeabilized with Cytofix/Cytoperm buffer (BD Bioscience). Next, the cells were intracellularly stained with anti-IFN-γ-PeCy7 (eBioscience), anti-IL-4-PE (eBioscience), and anti-IL-17-APC (eBioscience).

For Tfh cells analysis, splenocytes were surface stained with CD3e-V450 (eBioscience), CD4-FITC (eBioscience), CXCR5-APC (eBioscience), and PD-1-PE (eBioscience).

Intracellular ROS production in macrophages was determined by FCM using the DCFH-DA (Invitrogen) as a ROS-sensitive probe according to the standard procedure [19]. Briefly, the cells were collected, washed twice with PBS, and incubated in the dark with 10 μM DCFH-DA in serum-free DMEM at 37˚C for 30 min, followed by 2× wash in PBS.

                    

Mitochondrial membrane potential was detected using MitoProbe™ TMRM Kit for flow cytometry (Invitrogen). Briefly, J774A.1 cells were collected from culture, washed, and incubated with 20 nM TMRM reagent for 30 min at 37˚C, followed by 2× wash in PBS.

Mitochondrial ROS production was evaluated using MitoSox™ reagent (Life Technologies, Carlsbad, CA). Briefly, J774A.1 cells were rinsed with PBS and incubated with MitoSox™ for 10 min at 37˚C in 5% $CO_2$, followed by 2× wash in PBS.

Following staining or incubation as described above, cells were detected using a FACS Verse instrument (BD Bioscience) and analyzed with the FlowJo software (version 10.0.7, Tree Star, San Carlos, CA).

## Western blotting

Western blotting was carried out according to standard procedures. The J774A.1 cells were harvested and lysed in RIPA buffer (Invitrogen). The equal amounts of protein lysates were separated by 12% SDS-PAGE, transferred onto nitrocellulose membranes (Whatman, Maidstone, Kent, United Kingdom), and then detected with the following primary antibodies: anti-syk (Cat. No. ab3993, Abcam, Cambridge, UK), anti-p-syk (Cat. No. ab58575, Abcam), anti-Nox2 (Cat. No. ab80508, Abcam), anti-GAPDH (clone D16H11, Cell Signaling Technology/CST, Danvers, MA). HRP-conjugated anti-rabbit or anti-mouse IgG were used as secondary antibodies. Protein bands were detected using Immun-Star HRP Substrate (Bio-Rad) and digitally imaged using the Bio-Rad Gel Doc XR System (Bio-Rad).

## Statistical analysis

Statistical analysis was conducted using SPSS for Windows (version 26, IBM Corp.). The comparisons of two groups were analyzed by the Student's $t$-test, while multiple comparison tests were performed using one-way analysis of variance (ANOVA) followed by an LSD post hoc test. For all statistical analyses, $P < 0.05$ was considered significant.

## Results

### Increased ROS are accumulated in the livers of *S. japonicum*-infected mice

*S. japonicum* infection is known to cause granulomatous inflammation and thereby fibrosis in host liver [3]. *S. japonicum* infection was confirmed by the gross appearance and significantly increased weight of livers of mice at 8 weeks post-infection (Fig 1A), and subsequent data were collected. Infected mice showed severe liver immunopathology, including a typical granulomatous inflammation and fibrosis surrounding granulomas (Fig 1B and 1C). Meanwhile, significant accumulation of ROS was detected in the livers of infected mice with the fluorescent probe DCFH-DA, as demonstrated by an obvious green fluorescence comparing to normal liver showing a very weak fluorescence (Fig 1D). These data indicate that *S. japonicum* infection stimulates increased ROS accumulation in the mice livers.

### Blocking ROS production reduces hepatic immunopathology in *S. japonicum*-infected mice

We next examined whether the accumulation of ROS modulates the development of egg-induced hepatic immunopathology. A NOX inhibitor apocynin (Apo) was used to specifically inhibit NOX2-derived ROS production in *S. japonicum*-infected mice. Both liver and spleen showed a visible decrease in size in infected mice receiving Apo treatment (Fig 2A). Prominently decreased granulomatous responses, characterized by a smaller area of isolated hepatic granuloma in histological sections, were observed in the livers of infected mice with Apo

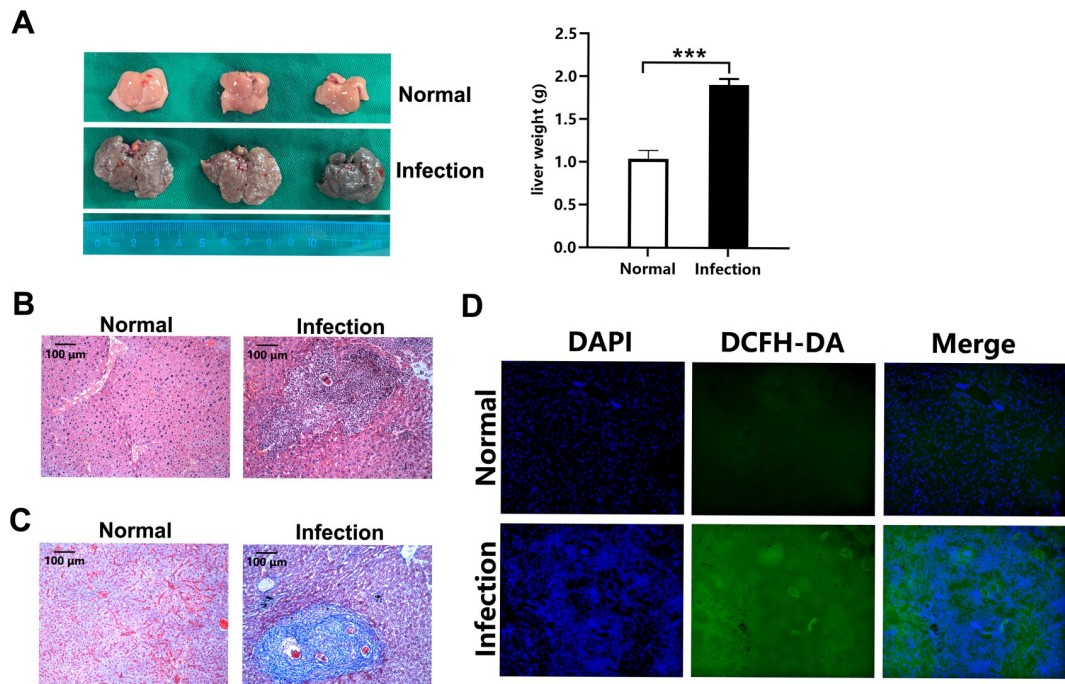

**Fig 1. Increased ROS are accumulated in the livers of *S. japonicum*-infected mice.** (A) Representative images of mouse livers from normal (uninfected) or *S. japonicum*-infected mice are shown. The bar graphs show the average liver weight (g) of mice. (B and C) Representative images of H&E (B) or Masson's trichrome (C) stained liver sections obtained from normal or infected mice are shown (original magnification, 100-fold; scale bar, 100 μm). (D) Immunofluorescence of frozen liver sections from normal or infected mice for nuclei using DAPI (blue) and ROS using DCFH-DA (green). (original magnification, 200-fold). Images and data expressed as the means ± SD of five mice per group are representative of three independent experiments. ***$P < 0.001$.

treatment (Fig 2B). In addition, Apo treatment significantly reduced the severity of liver fibrosis in infected mice, exhibiting a smaller fibrotic area and a lower collagen density (Fig 2C–2E). In addition, both ALT and AST levels were significantly reduced in infected mice with Apo treatment (Fig 2F). Taken together, these results indicate that the accumulation of ROS contributes to hepatic immunopathology in *S. japonicum*-infected mice.

## Blockade of ROS production inhibits M2 but promotes M1 macrophage differentiation in *S. japonicum*-infected mice

Macrophages are known as a major contributor for their capacity to produce a large amount of ROS upon various stimulations during hepatic inflammation and fibrosis induced by alcohol, lipid accumulation, endotoxins, and virus infections [12,20,21], but to date, there is no corresponding information on hepatic schistosomiasis. We observed an apparent increase of ROS production in macrophages from the livers of *S. japonicum* infected-mice using the fluorescent probe DCFH-DA (Fig 3A). We next investigated the involvement of the accumulation of ROS in macrophage differentiation during *S. japonicum* infection. Both CD16/32[+] M1 and CD206[+] M2 macrophages significantly increased in the liver of mice at 8 weeks post-infection with *S. japonicum*. Blocking ROS production using Apo in infected mice reduced the CD206[+] M2 macrophages, but meanwhile increased CD16/32[+] M1 macrophages (Fig 3B and 3C). These results suggest that the accumulation of ROS is critical for regulating macrophage differentiation during *S. japonicum* infection.

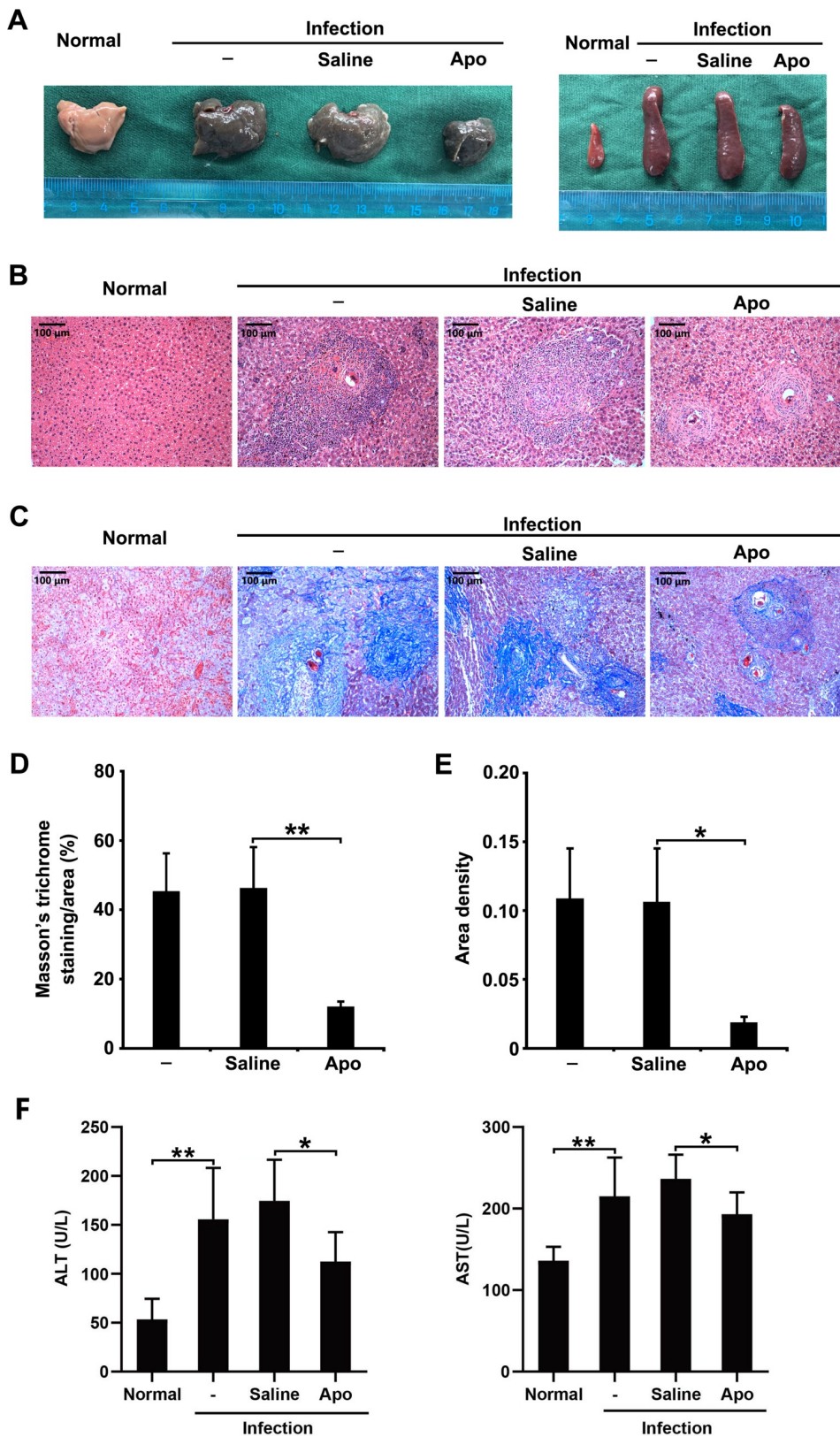

**Fig 2. Blocking ROS production reduces hepatic immunopathology in *S. japonicum*-infected mice.** *S. japonicum*-infected mice were treated with saline or NADPH oxidase inhibitor Apo. (A) Representative images of mouse livers and spleens are shown. Normal, uninfected mice. (B and C) Liver sections were H&E (B) or Masson's trichrome (C) stained (original magnification, 100-fold; scale bar, 100 µm). (D and E) Quantification of Masson's trichrome staining was performed using Image-Pro Plus software and is represented as the percentage of the stained-area per total area

(D) and area density (E). (F) Levels of alanine transaminase (ALT) and aspartate transaminase (AST) in the serum of mice were detected by an automatic biochemical analyzer. Data are expressed as the means ± SD of four mice per group and representative of three independent experiments. $^{*}P<0.05$, $^{**}P<0.01$.

## Blocking ROS production shifts from a Th2-type response towards a Th1-type response in *S. japonicum*-infected mice

Macrophage activation state also affects the way the Th cell response develops after activated antigen-presenting cells including macrophages migrate to lymph nodes and spleen. M1 macrophages promote a Th1 response but M2 macrophages promote a Th2 response [22–26]. We

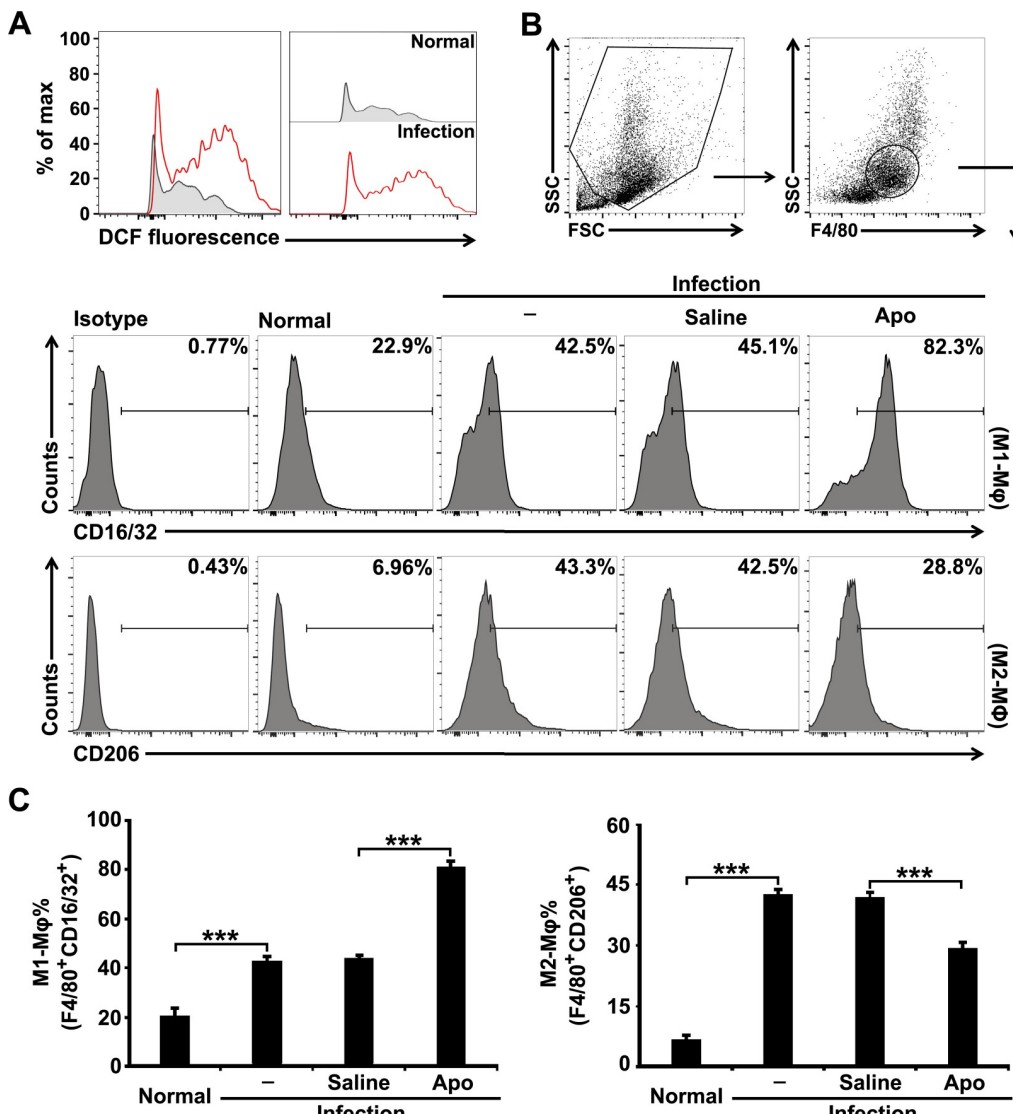

**Fig 3. Blocking ROS production inhibits M2 but promotes M1 macrophage differentiation in *S. japonicum*-infected mice.** (A) ROS production in macrophages from normal or infected mice was detected by FCM using the ROS-sensitive probe DCFH-DA. Normal, uninfected mice. (B and C) *S. japonicum*-infected mice were treated with saline or Apo. Expression of CD16/32 (M1) and CD206 (M2) on F4/80$^+$ macrophages were detected by FCM analysis. Representative histograms gated on F4/80$^+$ macrophages were shown (B). The bar graphs show the average percentages of F4/80$^+$CD16/32$^+$ M1 or F4/80$^+$CD206$^+$ M2 macrophages (C). Data are expressed as the means ± SD of four mice per group and representative of three independent experiments. $^{***}P<0.001$.

next examined whether ROS affect the Th1/Th2 dichotomy during *S. japonicum* infection. Following *S. japonicum* infection, the frequencies of Th1 (IFN-γ$^+$), Th2 (IL-4$^+$), Th17 (IL-17$^+$) (Fig 4A and 4B), and Tfh (CXCR5$^+$PD-1$^+$) cells (Fig 4C and 4D) in splenic CD3$^+$CD4$^+$ T cells apparently increased. Consistent with changes in macrophage differentiation, results showed decreased frequencies of Th2 cells (IL-4$^+$) but increased frequencies of Th1 (IFN-γ$^+$) in splenic CD4$^+$ T cells of *S. japonicum*-infected mice with Apo treatment(Fig 4B and 4C), whereas the frequencies of Th17 (IL-17$^+$) and Tfh (CXCR5$^+$PD-1$^+$) cells did not show any significant change (Fig 4). These results indicated that ROS also play an important role in enhancing Th2 responses during *S. japonicum* infection.

## SEA induce high production of ROS in macrophages to facilitate M2 macrophage differentiation

Schistosome eggs have been proven to preferentially promote M2 macrophage differentiation and induce M2 macrophage-rich granulomas during infection [4,6,7]. However, it is not clear whether stimulation of ROS production is necessary for *S. japonicum* egg-promoted M2 macrophage differentiation. Our results showed that a high level of ROS production was induced in murine macrophage J774A.1 after SEA stimulation (Fig 5A). Apo treatment significantly inhibited ROS production in SEA-stimulated J774A.1 (Fig 5A). FCM analysis further revealed that blocking ROS production with Apo reduced M2, but not M1, macrophage differentiation induced by SEA in J774A.1 cell culture (Fig 5B and 5C), suggesting that SEA stimulation leads to the accumulation of ROS and facilitate M2 macrophage differentiation.

## SEA stimulate ROS production in macrophages by NADPH oxidase and mitochondria

Studies have proven that the NADPH oxidase NOX2 is the important source of ROS in phagocytes. Syk-dependent NOX2 activation mediates ROS generation in macrophages [27–32]. Although we have confirmed that SEA induce a high level of ROS production in murine macrophages in Fig 5A, the underlying mechanism remains unknown. Thus, we used a specific small-molecule Syk inhibitor (R406) to examine whether SEA stimulate ROS generation via Syk-dependent NOX2 activation. Western blotting analysis revealed increased levels of phosphorylated Syk and NOX2 in murine macrophages J774A.1 after SEA treatment. In contrast, the ability of SEA to phosphorylate Syk and induce NOX2 expression was almost completely impaired after inhibition of Syk signaling (Fig 6A).

Being mitochondria another source of cellular ROS [33], we next examined the mitochondrial membrane potential as a parameter of mitochondria integrity and negatively associated with ROS production, using the TMRM probe, and mitochondrial ROS production using the MitoSOX probe. Following SEA treatment, mitochondrial membrane potential of J774A.1 was reduced (Fig 6B), which was paralleled by a moderate elevation of MitoSOX fluorescence in SEA-stimulated J774A.1 (Fig 6B). Taking together, these results indicate that SEA stimulate ROS production in macrophages by both NADPH oxidase and mitochondria.

## Discussion

Pathogens evoke rapid and massive production of ROS in the host, which play many important roles in the immune response and are intimately involved in host defense against invading pathogens, as well as immunopathological damage [34]. During hepatic schistosomiasis, considerable amounts of ROS are observed to accumulate in granulomas surrounding liver-trapped eggs, involved in triggering the development of schistosomiasis-associated liver

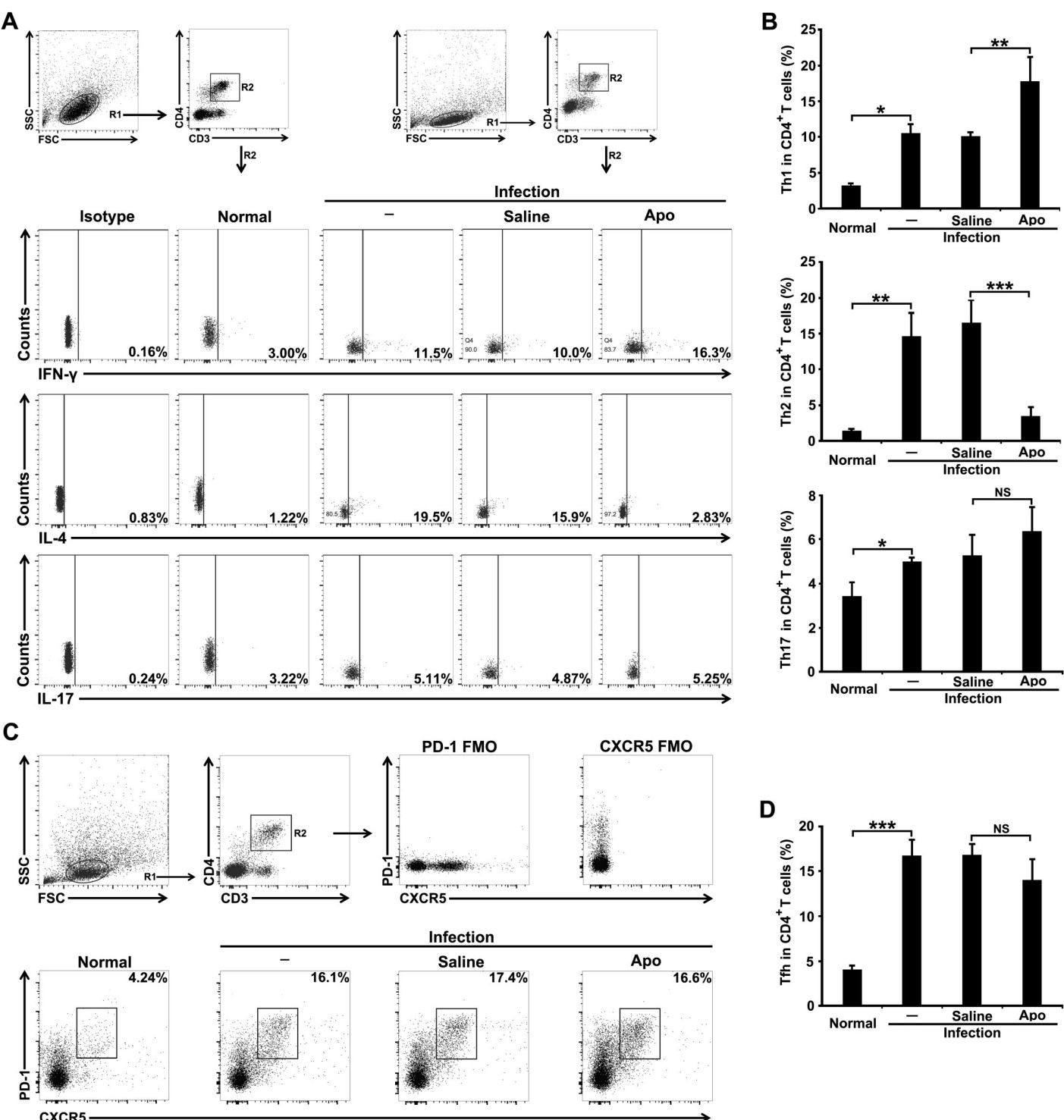

**Fig 4. Blocking ROS production shifts from a Th2-type response towards a Th1-type response in *S. japonicum*-infected mice.** *S. japonicum*-infected mice were untreated (-) or treated with saline or Apo orally. Normal, uninfected mice left untreated. (A and B) FCM gating strategies show total lymphocytes (R1) and CD3+CD4+ T cells (R2). Representative FCM dot plots show IFN-γ-(Th1), IL-4-(Th2), and IL-17-(Th17) producing CD3+CD4+ T cells from the livers of mice of each group (A). The bar graphs show the average percentages for Th1, Th2, and Th17 cells (B). (C and D) Representative FCM dot plots show CD3+CD4+CXCR5+PD-1+ Tfh cells from livers of mice of each group (C). The bar graphs show the average percentages for Tfh cells (D). Data are expressed as the means ± SD of four mice per group and representative of three independent experiments. *$P<0.05$, **$P<0.01$, ***$P<0.001$. NS indicating not significant.

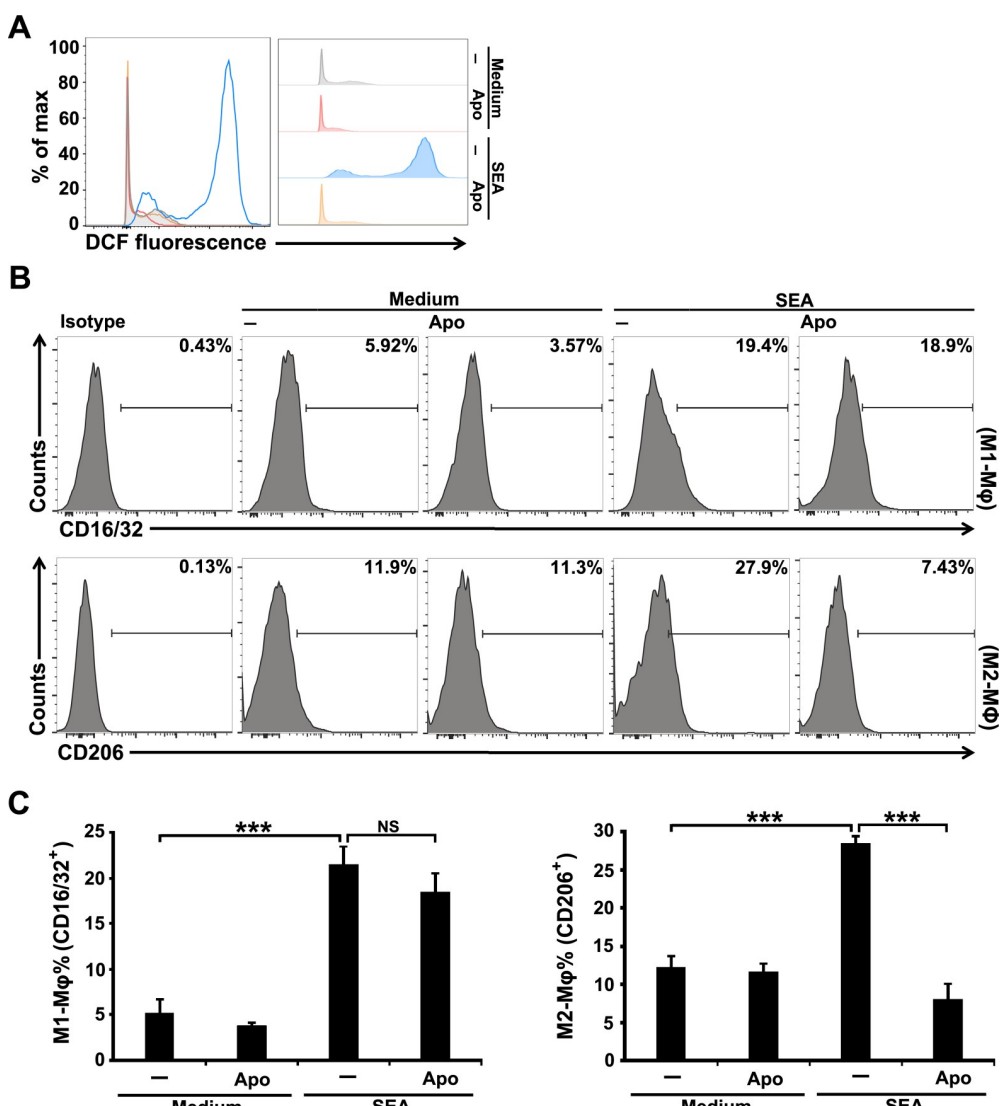

**Fig 5. SEA induce high production of ROS in macrophages to facilitate M2 macrophage differentiation.** J774A.1 cells were pretreated with Apo and then stimulated with 40 μg/m SEA for 24 hours. (A) ROS production in J774A.1 cells from each group was detected by FCM using the ROS-sensitive probe DCFH-DA. (B and C) Expression of CD16/32 (M1) and CD206 (M2) on J774A.1 cells were detected by FCM analysis. Representative histograms were shown (B). The bar graphs show the average percentages of CD16/32$^+$ M1 or CD206$^+$ M2 macrophages (C). Data are means ± SD of triplicate cultures and representative of three independent experiments. ***$P<0.001$. NS indicating not significant.

fibrosis [13,35]. However, whether schistosome eggs directly trigger the production of ROS, and if so, whether and how ROS promote granulomatous and fibrotic development in host liver is undefined. In this study, we illustrated that *S. japonicum* eggs evoke high production of ROS in macrophages, which is necessary for egg-mediated M2 macrophage differentiation and subsequently promoting granulomatous and fibrotic development in the liver of *S. japoni-cum*-infected mice.

Of note, ROS have been found to promote key events in the development of fibrotic diseases in important organs such as liver, lung, kidney, and heart by directly inducing profibrogenic factors and fibroblast activation or by indirectly initiating and accelerating inflammatory response [36,37]. During hepatic schistosomiasis, egg-induced granulomatous inflammation is

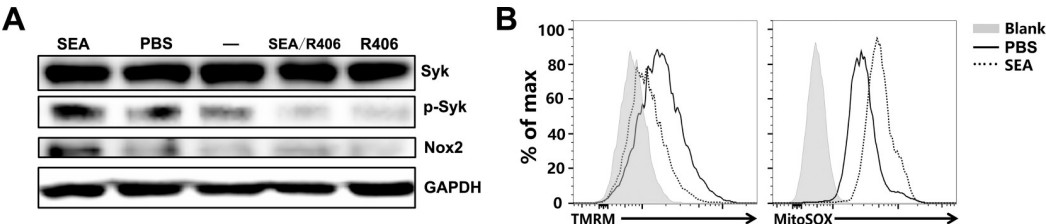

**Fig 6. SEA stimulate ROS production of macrophages by NADPH oxidase and mitochondria.** (A) J774A.1 cells were pretreated with Syk inhibitor R406 and then stimulated with PBS or 40 μg/m SEA for 24 hours. Cells were collected and whole cell lysates were extracted for detecting Syk, p-Syk, and Nox2 by immunoblotting. GAPDH was used as an internal control. (B) J774A.1 cells were stimulated with PBS or 40 μg/m SEA for 24 hours. Mitochondrial membrane potentials and ROS in cells were measured by FCM analysis using TMRM and MitoSOX, respectively. Data are representative of two independent experiments, each with triplicates.

the core factor leading to liver fibrosis [38]. However, it currently remains unclear whether ROS are involved in egg-induced hepatic inflammation and fibrogenesis. We showed here that *S. japonicum* eggs induced ROS play a critical role in promoting the development of hepatic fibrosis in mice with *S. japonicum* infection. Then, the precise mechanisms of how ROS regulate granulomatous inflammation to promote fibrosis during hepatic schistosomiasis become interesting and need to be elucidated.

Previous studies have demonstrated that increased ROS act as second messengers in macrophages and enhance fibrosis [39–41]. Schistosome eggs induce M2 macrophage-rich granulomas and Th2-biased immunity eventually leading to hepatic fibrosis [4,42]. Our current findings showed that the inhibition of ROS production in *S. japonicum*-infected mice reduced M2 macrophages and Th2 cells, but at the same time increased M1 macrophage and Th1 cells, concomitantly with alleviation of granulomatous inflammation and hepatic fibrosis. Thus, our *in vivo* findings suggest a previously unrecognized immunological role of ROS in enhancing M2 macrophage differentiation and Th2 responses, leading to liver immunopathological damage in hepatic schistosomiasis. Considering reciprocal regulation between macrophages and Th cells [22,23], ROS-enhanced M2 macrophages and Th2 cells are likely to have a relationship of mutual promotion. *In vitro* studies have proven that ROS production is necessary only for the differentiation of M2 but not M1 macrophages [14,15]. Similarly, our *in vitro* experiment also showed that inhibition of ROS production in macrophages only suppressed SEA-induced M2 macrophage differentiation but did not affect M1 macrophage differentiation. From these, we thus could infer that ROS are specifically required for schistosome egg-induced M2 macrophage differentiation in schistosomiasis, but the underlying mechanism about the different impacts of ROS on M1 macrophage differentiation between *in vivo* and *in vitro* needs to be further investigated.

Studies have suggested that the cellular sources of ROS are NADPH oxidase and/or mitochondria under both physiological and pathological conditions [27,43]. In response to various stimuli during chronic inflammatory oxidative stress, macrophages produce large amounts of ROS mainly through activation of NADPH oxidase NOX2 [31,32]. Consistently, we found NOX2-specific inhibitor (Apo) efficiently inhibited SEA-stimulated ROS production in macrophages, suggesting NOX may act as the main source of ROS production in SEA-stimulated macrophages. In addition, our *in vitro* data further demonstrated that SEA stimulated ROS generation in macrophages via Syk-dependent activation of NOX2. Low-density lipoprotein was previously shown to induces ROS generation in macrophages through sequential activation of TLR4, Syk, PLCγ1, PKC, and NOX2 [30]. Given schistosome egg-derived antigens have been proven to signal through TLR4 [44,45], it is highly likely that SEA induce ROS generation

through a similar mechanism, but still requiring further investigation. Besides, we found that SEA stimulated the generation of ROS in mitochondria, though the production was not that high as by NOX. This may also serve to explain, at least in part, the massive distribution of ROS surrounding deposited eggs in the livers of *S. japonicum*-infected mice. However, the precise mechanisms and respective contribution involved in these two pathways for SEA-stimulated ROS generation remain unclear. Future studies, e.g., deeper investigation of the roles and mechanisms of NADPH oxidase- or mitochondria-derived ROS production in schistosomiasis, should be conducted.

In summary, our study shows that ROS accumulation in the liver of *S. japonicum*-infected mice contributes to hepatic immunopathology and further demonstrates that schistosome eggs induce ROS generation to promote M2 macrophage differentiation. Our results provide a target regarding schistosome eggs-induced ROS production, which can be manipulated to regulate immunopathology in hepatic schistosomiasis.

## Author Contributions

**Conceptualization:** Yanxiong Yu, Junling Wang, Xiaojun Chen, Jifeng Zhu, Sha Zhou, Chuan Su.

**Data curation:** Yanxiong Yu, Junling Wang, Sha Zhou.

**Funding acquisition:** Sha Zhou, Chuan Su.

**Investigation:** Yanxiong Yu, Junling Wang, Xiaohong Wang, Pan Gu, Zhigang Lei, Rui Tang, Chuan Wei, Lei Xu, Chun Wang, Ying Chen, Yanan Pu, Xin Qi, Beibei Yu, Xiaojun Chen, Jifeng Zhu, Yalin Li, Sha Zhou.

**Methodology:** Yanxiong Yu, Junling Wang, Xiaohong Wang, Pan Gu, Zhigang Lei, Rui Tang, Chuan Wei, Lei Xu, Chun Wang, Ying Chen, Yanan Pu, Xin Qi, Beibei Yu, Xiaojun Chen, Jifeng Zhu, Yalin Li, Zhijie Zhang, Sha Zhou.

**Project administration:** Chuan Su.

**Supervision:** Sha Zhou, Chuan Su.

**Writing – original draft:** Yanxiong Yu, Junling Wang, Sha Zhou.

**Writing – review & editing:** Sha Zhou, Chuan Su.

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
