## [Decision Letter · Decision Letter 0]

16 May 2021

Dear Dr. Chuan Su,

Thank you very much for submitting your manuscript "Schistosome eggs stimulate reactive oxygen species production to promote hepatic pathology in schistosomiasis through enhancing alternatively activated macrophage differentiation" for consideration at PLOS Neglected Tropical Diseases. As with all papers reviewed by the journal, your manuscript was reviewed by members of the editorial board and by several independent reviewers. 

The reviewers felt the paper was interesting and very well written. Based on the reviews, we are likely to accept this manuscript for publication, providing that you modify the manuscript according to the review recommendations. I would particularly ask you to consider comments made by reviewer 2 who suggests that some data regarding T-cell - macrophage interactions could be interpreted differently. The final interpretation is up to you.

Sincerely,

John Pius Dalton, PhD

Associate Editor

Sergio Oliveira

Deputy Editor

Reviewer's Responses to Questions

**Key Review Criteria Required for Acceptance?**

**Methods**

-Are the objectives of the study clearly articulated with a clear testable hypothesis stated?

-Is the study design appropriate to address the stated objectives?

-Is the population clearly described and appropriate for the hypothesis being tested?

-Is the sample size sufficient to ensure adequate power to address the hypothesis being tested?

-Were correct statistical analysis used to support conclusions?

-Are there concerns about ethical or regulatory requirements being met?

Reviewer #1: (No Response)

Reviewer #2: (No Response)

**Results**

-Does the analysis presented match the analysis plan?

-Are the results clearly and completely presented?

-Are the figures (Tables, Images) of sufficient quality for clarity?

Reviewer #1: (No Response)

Reviewer #2: (No Response)

**Conclusions**

-Are the conclusions supported by the data presented?

-Are the limitations of analysis clearly described?

-Do the authors discuss how these data can be helpful to advance our understanding of the topic under study?

-Is public health relevance addressed?

Reviewer #1: (No Response)

Reviewer #2: (No Response)

**Editorial and Data Presentation Modifications?**

Reviewer #1: (No Response)

Reviewer #2: (No Response)

**Summary and General Comments**

Reviewer #1: Interesting study on ROS and hepatic schistosomiasis. Experiments were comprehensive, performed well, and results presented effectively.

Figure 1 (and others), All panels need a scale bar. Panels B and C do not need the staining type labelled, this information only needs to be in the legend. While the images of livers were informative the use of weights would have been more precise and could have enabled statistical comparison.

Figure 4 please clarify the four groups especially the difference between “-“ and “saline” groups. Does “-“ indicate no oral dosage at all?

Line 66 “(S. mansoni)” is not needed and further species only require the abbreviated genus name ie delete “Schistosoma japonicum”.

Line 80 please add reference.

What lobe of the liver was used for histology?

Can you confirm that LSP was demonstrated as LPS free?

Line 242 (Figure 5C-5E) should be fig 2? Similarly on line 273?

Discussion was well written and was effectively supported by appropriate references. Line 375 it would be nice for the authors to propose future experiments to expand their findings below “….need

to be further explored.”.

Reviewer #2: This manuscript describes an investigation into the association between ROS production, macrophage differentiation and the development of hepatic pathology during infection with S. japonicum. The major finding from this study is that a significant increase in ROS in the liver of infected mice is causally linked to the formation of granulomas, as the administration of a specific inhibitor of NOX reduced pathology. In addition, the presence of this inhibitor also impacted the developing immune response in animals, reducing the differentiation of M2-type macrophages and increasing the population of M1 cells in the liver, and reducing the number of IL-4 secreting T cells in the spleen. This was partly repeated in vitro, where the presence of the inhibitor prevented the SEA induced differentiation of M2 like cells, but unlike the scenario in vivo, did not enhance the development of M1. The in vitro analysis also attempted to shed some light on the possible mechanism by which the SEA drives ROS-mediated differentiation of macrophages, with data suggesting a Syk-dependent activation of NOX. The outcomes contribute to improving understanding of the progression of liver pathology after infection with Schistosoma and how the parasite may be influencing the development of specific macrophage phenotypes.

The study utilised standard protocols which were performed well and interpreted correctly. Although, it should be clarified in text whether apocynin is a specific inhibitor of NOX-2.

However, while the data has shown that SEA induces macrophage differentiation at least in part via ROS production and that the ROS in the liver contributes to the pathology, the data does not show (as suggested by the authors) that the macrophages then contribute to the granuloma formation. I would suggest that this conclusion (Line 377) is re-considered. In addition, there is no evidence that the CD206+ macrophages are driving the differentiation of IL-4 secreting T-cells; it may in fact equally be the other way around and the IL-4 produced by T-cells is driving the differentiation of macrophages in vivo. While the data is presented accurately, the conclusions drawn need to be reconsidered – as the experimental design does not support the demonstration of causal sequential cascade of SEA – ROS- M2 - granuloma/Th2.

Some minor comments:

1. Introduction Line 78: M2 macrophages are not phenotypically the same as alternatively activated macrophages, which have been characterised specifically as IL-4 activated macs. Please refer to the recent literature around the nomenclature of macrophages

2. Results Line 220: I would suggest replacing “Massive” with more scientific terminology

3. Some of the figure numbers are incorrectly listed throughout the results text

4. Fig 1: scale bars should be added to the histology figures and described in the legend.

5. Fig 1: I am not convinced by the DCFH-DA staining which is very diffuse and almost non-specific. It may help the interpretation of this data if the tissue can be counterstained for some other liver/granuloma structure – or even DAPI at the most basic.

6. Fig 2: It would be of value here to have a measure of liver health to accompany the assessment of pathology in the liver (perhaps serum ALT or equivalent). While the livers certainly seem smaller in the animals treated with APO, they do not look any healthier.

7. Line 276; as mentioned the data does not show that macrophage activation is subsequently controlling the Th1/Th2 response; it does show that both CD206+ macrophages and Th2 responses are present in infected animals and reduced in animal given Apo.

8. Fig 6: The sample size should be added to the legend for this figure

9. There are a number of language and typo errors throughout.

PLOS authors have the option to publish the peer review history of their article (what does this mean?). If published, this will include your full peer review and any attached files.

Reviewer #1: No

Reviewer #2: No

Figure Files:

Data Requirements:

Reproducibility:

References

---

## [Editor Report · Decision Letter 1]

31 Jul 2021

Dear Dr. Chuan Su,

We are pleased to inform you that your manuscript 'Schistosome eggs stimulate reactive oxygen species production to enhance M2 macrophage differentiation and promote hepatic pathology in schistosomiasis' has been provisionally accepted for publication in PLOS Neglected Tropical Diseases.

Best regards,

John Pius Dalton, PhD

Associate Editor

Sergio Oliveira

Deputy Editor

Paper is well performed and very interesting. Thnak you for responding to comments and adding new data.

---

## [Editor Report · Acceptance letter]

11 Aug 2021

Dear Prof. Su,

We are delighted to inform you that your manuscript, "Schistosome eggs stimulate reactive oxygen species production to enhance M2 macrophage differentiation and promote hepatic pathology in schistosomiasis," has been formally accepted for publication in PLOS Neglected Tropical Diseases.

Best regards,

Shaden Kamhawi

co-Editor-in-Chief

Paul Brindley

co-Editor-in-Chief
